# Spatiotemporal Evolution of Landscape Ecological Risk Based on Geomorphological Regionalization during 1980–2017: A Case Study of Shaanxi Province, China

**Di Liu [1,2]**, **Hai Chen [1,2,*]**, **Hang Zhang [1,2]**, **Tianwei Geng [1,2]** and **Qinqin Shi [1,2]**

[1]  College of Urban and Environmental Science, Northwest University, Xi'an 710127, China;
     lcx@stumail.nwu.edu.cn (D.L.); zhrwdl2000@126.com (H.Z.); gengtianwei1002@126.com (T.G.);
     sqq116@stumail.nwu.edu.cn (Q.S.)
[2]  Shaanxi Key Laboratory of Earth Surface System and Environmental Carrying Capacity, Xi'an 710127, China
*   Correspondence: chw@nwu.edu.cn; Tel.: +86-029-88308427

**Abstract:** Land surface elements, such as land use, are in constant change and dynamically balanced, driving changes in global ecological processes and forming the regional differentiation of surface landscapes, which causes many ecological risks under multiple sources of stress. The landscape pattern index can quickly identify the disturbance caused by the vulnerability of the ecosystem itself, thus providing an effective method to support the spatial heterogeneity of landscape ecological risk. A landscape ecological risk model based on the degree of interference and fragility was constructed and spatiotemporal differentiation of risk between 1980 and 2017 in Shaanxi Province was analyzed. The spatiotemporal migration of risk was demonstrated from the perspective of geomorphological regionalization and risk gravity. Several conclusions were drawn: The risk of Shaanxi Province first increased and then decreased, at the same time, the spatial differentiation of landscape ecological risk was very significant. The ecological risk presented a significant positive correlation but the degree of autocorrelation decreased. The risk of the Qinba Mountains was low and the risk of the Guanzhong Plain and Han River basin was high. The risk of Loess Plateau and sandstorm transition zone decreased greatly and their risk gravities shifted to the southwest. The gravity of the Guanzhong Plain and Qinling Mountains had a northward trend, while the gravity of the Han River basin and Daba Mountains shifted to the southeast. In the analysis of typical regions, there were different relationships between morphological indicators and risk indexes under different geomorphological features. The appropriate engineering measures and landscape management for different geomorphological regionalization were suggested for effective reduction of ecological risks.

**Keywords:** landscape ecological risk; geomorphological regionalization; risk gravity; human interference; Shaanxi Province

---

## 1. Introduction

Land use is a comprehensive reflection of human economic and social activities acting on terrestrial surface resources and the natural environment. It is the result of the combined action of human activities and natural processes [1]. As the most direct manifestation of human utilization of the natural environment [2], land use and other land surface elements (water, soil, atmosphere, biology, etc.) are in constant change and have a dynamic balance; thus, these processes drive the changes in global ecological processes and form the spatial differentiation of surface landscapes. The beginning of the "Anthropocene" is reflected by changes in global landscape patterns and ecological functions [3]

and this new period is related to the multi-directional evolution of the ecological environment and has generated many ecological risks. The assessment and prevention of ecological risk has become a popular research topic in geography and ecology in response to the comprehensive management of social ecosystems [4–8].

Ecological risk is the possibility of adverse ecological effects due to the external pressures exerted on ecosystem components [9–11]. Ecological risk assessment (ERA) started in the 1980s and initially focused on the impact of chemical pollutants on human health [6,12]. Subsequently, due to the complexity of risk sources and the diversity of risk receptors, ERA has gradually extended to the category of socio-economic-natural complex ecosystems, including multiple risk sources and risk receptors [13]. The current research methods for assessing ecological risk include the perspective of risk source-sink and landscape structure [14,15]. Risk assessment based on source-sink can be used to identify risk sources and receptors in the area threatened by an exposure-response [16,17], and additional methods include the PSR framework [18] and the RRM model [19–21]. Risk assessment based on the landscape structure relies on the coupling and correlation perspective of patterns and processes and focuses more attention on the spatiotemporal heterogeneity of risks; this approach often uses the land use intensity parameters [2,22,23] and landscape loss degree [8,24,25] as the main elements. In the absence of ecological monitoring data, the landscape pattern characteristics reflected by land use can clarify the impact of quantitative change and spatial allocation of land use on the ecosystems and processes to reflect the spatial differentiation of the ecological environment [26–28]. Therefore, using the landscape of heterogeneous land units composed of different types of ecosystems as the perspective of evaluation, this paper explores the impact of human activities on land use based on the landscape pattern, providing the best generalized approach for further clarifying the evolutionary mechanism of ecological risks [5]. This method has been applied in many areas, such as watersheds [7,29], coastal zones [30] and administrative regions [25].

Land uses have spatiotemporal heterogeneity and are subject to regional topographic and geomorphic characteristics [31], which are associated with spatiotemporal changes in landscape pattern and ecological risks. This spatiotemporal variation is mainly manifested as the directional migration of ecological risks on terrain with different gradients [8]. Correlation studies show that the simple response of ecological risk to terrain is achieved using the methods of ecological risk distribution in the terrain grade area ratio [22,32], ecological risk index calculation on terrain grade [23], and terrain and ecological risk section line [8,33]. However, based on the unique and single geomorphologic features in a study area, it is difficult to discuss the influence of geomorphologic differentiation on ecological risk evolution in a large region. Therefore, this method may be an effective way to identify the variation trend of ecological risk and its related elements from the perspective of geomorphological regionalization.

Using Shaanxi Province, China, which has various geomorphic types, as an example, this paper aims to restore and explain the causal chain of "geomorphologic features-human drivers-pattern change-risk response" under the ecological risk paradigm of landscape pattern. At the same time, the spatiotemporal evolution model of risk is explored based on the perspective of geomorphological regionalization to promote the transition of regional ecosystems from vulnerable to sustainable through ecological restoration and landscape pattern optimization.

## 2. Study Area and Data Sources

### 2.1. Study Area

Shaanxi Province (105°29′ E–111°15′ E, 31°42′ N–39°35′ N) is located in central China (Figure 1). The administrative divisions include 10 cities, which account for a total catchment area of $20.57 \times 10^4$ km$^2$. The average annual temperature of the entire province is 13.7 °C, and temperature decreases from south to north. The average annual precipitation is 680 mm, with more precipitation in the south and less precipitation in the north. Southern Shaanxi Province has a humid climate, Guanzhong Plain has a semi-humid climate, and northern Shaanxi Province has a semi-arid

climate [34,35]. The Qinba Mountains in southern Shaanxi are adjacent to the Qinling Mountains in the north and the Daba Mountains in the south, and the Han River basin in located in the central part. The Qinba Mountains are an important water source in the South-north Water Diversion Project and an ecological function-restricted development zone. There are serious landslides, floods and other geological disasters in this mountainous area. The central part of the province is the Guanzhong Plain, which is high in the west and low in the east, high in the north and south, and low in the middle. The Guanzhong Plain is a major urban agglomeration area with a strong intensity of human activity. The geomorphologic types of northern Shaanxi include the sandstorm transition zone and the Loess Plateau (including tableland, beam, and gully) geomorphology. Precipitation is scarce and unevenly distributed in this region. It is a typical ecologically fragile region in the world, with a high drought frequency and serious soil erosion. The spatial heterogeneity of human disturbances caused by the various geomorphic differentiation provides an ideal case study for discussing the spatiotemporal differentiation of eco-logical risk from the perspective of geomorphological regionalization.

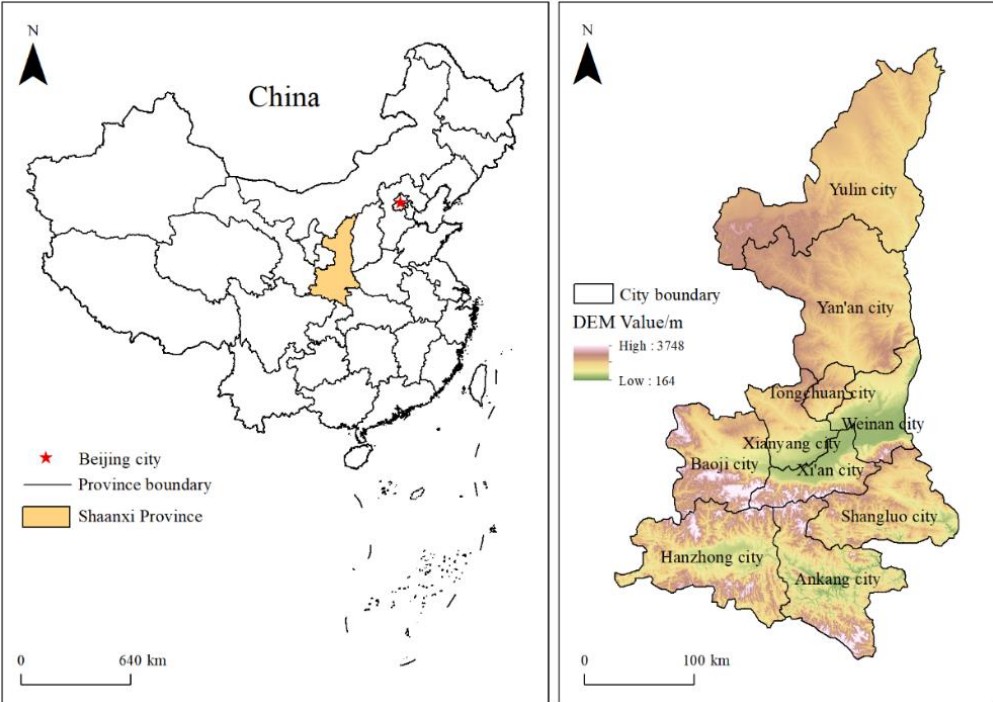

**Figure 1.** Location and DEM of Shaanxi Province, China.

*2.2. Data Sources*

The land use data of Shaanxi Province in 1980, 2000 and 2017 were derived from the 1:100,000 vector data set of land use provided by the Cold and Arid Regions Sciences Data Center of China (http://westdc.westgis.ac.cn). These data were based on Landsat MSS, TM and ETM images, which were generated by combining visual interpretation of terrain and cognition of image spectrum. The accuracy of data interpretation was 95%, which was the highest precision land use data product in China. According to the classification standard of land use of China (GB/T21010-2017), land use types were divided into six categories: farmland, forest, grassland, water body, construction land, and unused land. Land use data were resampled to 30-m grids for subsequent analysis. The digital elevation model (DEM) was constructed from data sourced from the Geospatial Data Cloud (http://www.gscloud.cn/), and the elevation of the study area was extracted. The boundary of geomorphological regionalization was vectorized from the geographic atlas of Shaanxi Province, and the subarea types included the sandstorm transition zone, Loess Plateau, Guanzhong Plain, Qinling Mountains, Han River basin and Daba Mountains. Then, the boundary vector files were generated.

## 3. Research Methods

### 3.1. The Spatial Display Approach

Regarding ecological risk spatialization, the equidistant sampling grid method was the most common type [15]. In landscape ecology, when the area of a landscape sample (that is, the sampling unit) is 2–5 times the area of a landscape type patches, it can comprehensively reflect the landscape pattern information around the sampling point [36,37]. Based on this criterion, a grid size of 12 km was selected as a suitable scale for the spatialization of ecological risks. The study regions were divided into 1471 cell grids and the risk values were assigned to the corresponding risk point sets. As a regional variable, the ecological risk index can be measured by spatial evolution trends using the geostatistical semi-variance function [28,30]. The formula was given as follows:

$$\gamma(h) = \frac{1}{2N(h)} \sum_{i=1}^{N(h)} [Z(x_{i+h}) - Z(x_i)]^2 \qquad (1)$$

where $\gamma(h)$ is the semi-variance function; $h$ is the step length; $Z(x_{i+h})$ and $Z(x_i)$ are the observed value of the ecological risk index at spatial location $x_i$ and $x_{i+h}$, respectively; and $N(h)$ is the number of samples within the interval distance $h$.

In this paper, GS$^+$7.0 software was used to achieve the best fit of the semi-variance function [30]. Combined with the model fitting effect, the spatial analysis of ecological risk in 1980 was based on the spherical model for fitting and the spatial analysis of ecological risk in 2000 and 2017 was based on the exponential model. Finally, a spatial distribution map of the ecological risk index was created by ordinary Kriging interpolation using the Geostatistics Analyst Tools of ArcGIS 10.2.

### 3.2. Landscape Index Method

The landscape index model with the landscape disturbance indexes and the landscape vulnerability indexes coupled with the area weighting function as the main components method is widely used to quantify the characteristics of landscape patterns, providing an effective method to clarify the spatial heterogeneity of landscape ecological risk; this method is the main approach of landscape ERA [38,39]. In this paper, the ecological risk index model was established to describe the relative value of the ecological risk index in the evaluation unit. The formula is as follows:

$$ERI_k = \sum_{i=1}^{n} \frac{S_{ki}}{S_k} \sqrt{E_i \times F_i} \qquad (2)$$

where $ERI_k$ denotes the landscape ecological risk index of an unit $k$; $n$ is the number of landscape types; $E_i$ and $F_i$ are the disturbance indexes and fragility indexes of landscape $i$, respectively; $S_{ki}$ is the area of landscape component $i$ in unit $k$; and $S_k$ is the total area of unit $k$. By referring to previous studies, the detailed calculation methods of $E_i$ are given in Table 1 [40].

Landscape fragility represented the vulnerability of the ecosystem structure represented by each land use type. The level of vulnerability can reflect the sensitivity and resistance of the landscape to interference by external risk. The greater the vulnerability of a landscape is, the greater the ecological risk is, and vice versa. By referring to relevant studies [8,41] and using the expert scoring method, the fragility of land use was divided into 6 grades (Table 2).

**Table 1.** Calculation of landscape disturbance indexes.

| Landscape Disturbance Indexes | Calculation Method | Meaning |
|---|---|---|
| Landscape fragmentation index ($C_i$) | $C_i = n_i / A_i$ | $C_i$ refers to the process in which landscape types change from single, homogeneous, and continuous to complex, heterogeneous and discontinuous due to the interference of nature or human beings. The increase in $C_i$ indicates a decrease in system stability and an aggravation of ecological risk. |
| Landscape isolation index ($N_i$) | $N_i = \frac{A}{2A_i} \sqrt{\frac{n_i}{A}}$ | $N_i$ refers to the degree of separation of different patch numbers in landscape types. The increase in the $N_i$ index indicates that the landscape is more scattered, the landscape distribution is more complex, and the fragmentation degree is higher. |
| Landscape dominant index ($DO_i$) | $DO_i = \frac{Q_i + M_i}{4} + \frac{L_i}{2}$ | $DO_i$ is an indicator used to measure the importance of patches in a landscape, which directly reflects the impact of this landscape type on landscape pattern. |
| Landscape disturbance index ($E_i$) | $E_i = aC_i + bN_i + cD_i$ | $E_i$ reflects the disturbance degree of an ecosystem represented by different landscape types. |

Note: $n_i$ is the number of patches in the $i$th landscape; $A_i$ is the area of the $i$th landscape; $A$ is the landscape total area; $Q_i$ is the sample number of the $i$th pattern/total number of samples; $M_i$ is the number of $i$th patches/total number of patches; $L_i$ is the area of the $i$th patches/area of samples; $a$, $b$ and $c$ are the weighted coefficients, and $a + b + c = 1$. As suggested in the literature [28,30], $a = 0.5$, $b = 0.3$ and $c = 0.2$ in this study.

**Table 2.** The grades of fragility and corresponding values of fragility for the different land use types.

| Land Use Types | Grades of Fragility | Values of Fragility |
|---|---|---|
| Construction land | 1 | 0.0476 |
| Forest | 2 | 0.0952 |
| Grassland | 3 | 0.1429 |
| Farmland | 4 | 0.1905 |
| Waterbody | 5 | 0.2381 |
| Unused land | 6 | 0.2857 |

### 3.3. Spatial Autocorrelation Method

Spatial autocorrelation is the basic form of spatial dependence, and this value reflects the degree of spatial correlation of the risk variables. Global spatial autocorrelation is used to measure the global spatial distribution characteristics of risk value. The Moran's *I* index can measure the risk value from the degree of global spatial correlation and spatial difference [2,42]. The formula is as follows:

$$I(d) = \frac{n \sum\limits_{i=1}^{n} \sum\limits_{j=1}^{n} w_{ij}(x_i - \overline{x})(x_j - \overline{x})}{\sum\limits_{i=1}^{n} (x_i - \overline{x}) \sum\limits_{i=1}^{n} \sum\limits_{j=1}^{n} w_{ij}} \tag{3}$$

where $x_i$ and $x_j$ are the values of variable $x$ at adjacent space points, respectively; $\overline{x}$ is the average of the variables; $w_{ij}$ is the adjacent weight; and $N$ is the total number of risk points. The value of Moran's *I* index is [−1, 1]. When Moran's *I* < 0, there is a negative correlation. When Moran's *I* > 0, there is a positive correlation. When Moran's *I* = 0, there is no correlation. Finally, the autocorrelation is stronger as the absolute value approaches 1.

Local anomalies can be introduced into the local spatial correlation index LISA to identify the spatial aggregation pattern of cold spots and hot spots. The formula is as follows:

$$I_i = Z_i \sum_{j=1}^{n} w_{ij} Z_j, (i \neq j) \tag{4}$$

where $I_i$ is the unit value of the LISA index, $Z_i$ and $Z_j$ are the standardized risk values of units $i$ and $j$, respectively, and $w_{ij}$ is the spatial weight matrix.

In this paper, GeoDA software was used to measure the spatial distribution pattern of the ecological risk index in three periods in Shaanxi Province. A Moran's *I* scatter plot was generated, and the Moran's *I* values and *Z* values of the standardized statistics were extracted ($p = 0.01$).

*3.4. Standard Deviational Ellipse and Risk Gravity*

The standard deviation ellipse (SDE) is widely used in the field of spatial statistics, and it is a common method used to quantitatively analyze the overall characteristics of the spatial distribution of point elements. As spatial variables, the global characteristics of the spatial distribution of risk points are presented from multiple angles using the SDE and its gravity tool [27,43]. The expression of gravity is as follows:

$$X = \frac{\sum_{i=1}^{n} w_i x_i}{\sum_{i=1}^{n} w_i}; Y = \frac{\sum_{i=1}^{n} w_i y_i}{\sum_{i=1}^{n} w_i} \tag{5}$$

where $(X, Y)$ is the weighted average gravity, and $(x_i, y_i)$ represents the coordinates of risk points. $w_i$ represents the weight. In this paper, the direction distribution module of ArcGIS10.2 was used to calculate the risk gravity of geomorphological regionalization, with the risk value as the weight, and to visualize the space.

## 4. Results

*4.1. Spatial and Temporal Differentiation of Landscape Ecological Risk*

The ecological risk value was classified using the ArcGIS10.2 natural breakpoint method [44]: grade I (ERI < 0.1190), grade II (0.1190 ≤ ERI < 0.1271), grade III (0.1271 ≤ ERI < 0.1353), grade IV (0.1353 ≤ ERI < 0.1445), and grade V (ERI > 0.1445); this method enabled us to generate the spatial distribution of ecological risk in Shaanxi Province from 1980 to 2017 (Figure 2). The landscape ecological risk index of Shaanxi Province was 0.1282, 0.1287 and 0.1273, respectively, and the risk value first increased and then decreased, with an overall decrease of 0.78%. The spatiotemporal differentiation of ecological risk was very significant in Shaanxi Province. An analysis of Figure 2 shows that the grade I-II risk area in 1980 accounted for approximately half of the study area, and this area was mainly distributed in the Qinba Mountains and the southern part of the Loess Plateau. The grade IV-V risk area accounted for 19.25% of the study area, and this area was mainly distributed in the northern Loess Plateau, Guanzhong Plain and Han River basin. Compared with 1980, the grade IV risk area in the northern part of the Loess Plateau increased significantly. At the same time, the grade V risk in the Guanzhong Plain extended to the north and south. The urban built-up areas of the Han River basin formed the core of the grade V risk area. Compared with 2000, the ecological risk in 2017 decreased significantly. The grade IV risk area in the north of the Loess Plateau was cut off by the grade III risk area. The grade V risk area in the south of Xi'an has been cut off, which formed a double core area of high risk in Xianyang city and Weinan city. The grade IV-V risk area of the Han River basin has been greatly reduced, and the grade V risk area has been withdrawn from the Hanzhong urban construction area.

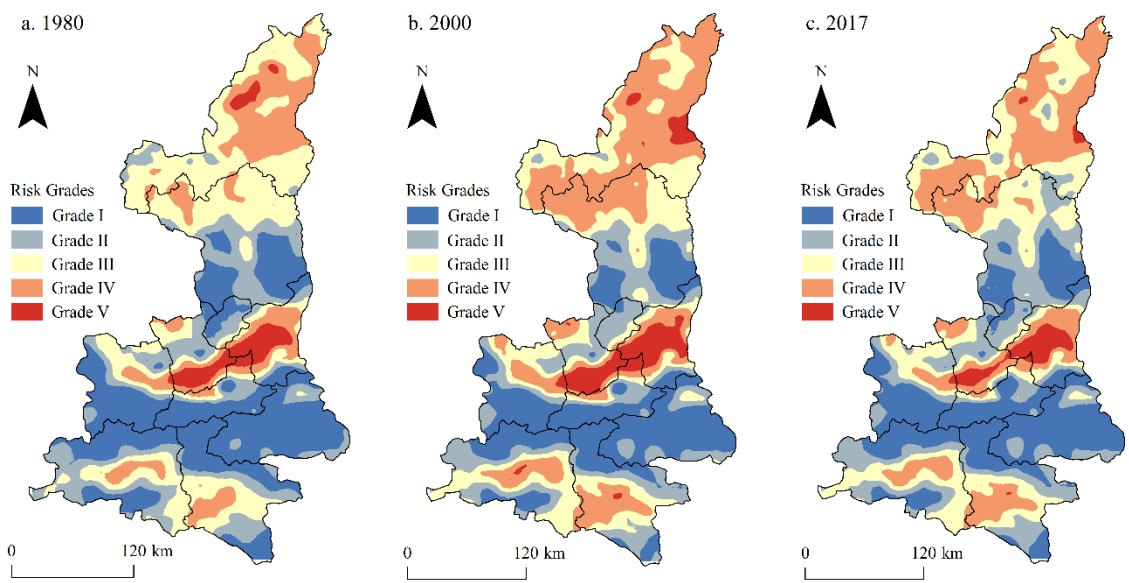

**Figure 2.** Spatial distribution of landscape ecological risk in Shaanxi Province during 1980–2017.

*4.2. Spatial Autocorrelation Analysis*

From 1980 to 2017, the Moran's *I* index of ecological risk in Shaanxi Province was 0.7279, 0.7277 and 0.7053, respectively, and the Z-values of the random distribution test were 37.9032, 38.2940 and 35.5271, respectively. This finding indicates that the ecological risk presented a significant positive correlation. The Moran's *I* index showed a downward trend over time, indicating that the spatial autocorrelation degree of ecological risk had been reduced. The landscape pattern and ecological process were affected by the evolution of land use types, and these changes affected the landscape ecological risk, leading to the gradual transition of its spatial distribution from aggregation to uniformity.

Based on the LISA index, GeoDA software was used to analyze the clustering characteristics of landscape ecological risk in Shaanxi Province (Figure 3), and the result was significant at the $p \leq 0.05$ level. There is a large number of "high–high" and "low–low" agglomeration areas in Figure 3a. Risk hotspots were mainly concentrated in the north of the Loess Plateau, the sandstorm transition zone and the central Guanzhong Plain. The risk hotspots in 2000 showed an increasing trend, especially in the Loess Plateau. However, the area of risk hotspots in this region decreased to some extent in 2017, and the significance of spatial agglomeration weakened. The risk cold spots were mainly located in the Qinling Mountains and the eastern region of Yan'an, and there was little change in the risk cold spots during the study period, which was closely related to the fact that most of the cold spot areas were located in semi-natural and natural areas. In addition, there were some spatial "singularity spots" in the research region. The "low–high" spots were located in the periphery of risk hotspots, and these sites were greatly disturbed by human activities and had farmland that was easily converted into other land use types. The change in land use types affected the habitat vulnerability and affected the regional landscape pattern, which made it more possible for this part of the region to transform into high risk. The "high–low" spots were located around the risk cold spots, which may be caused by the unused land that was surrounded by forest and grassland. In general, the singularity spots were located in the transitional region of land use change and landscape pattern evolution and they were also the areas where ecological risk was most likely to spread.

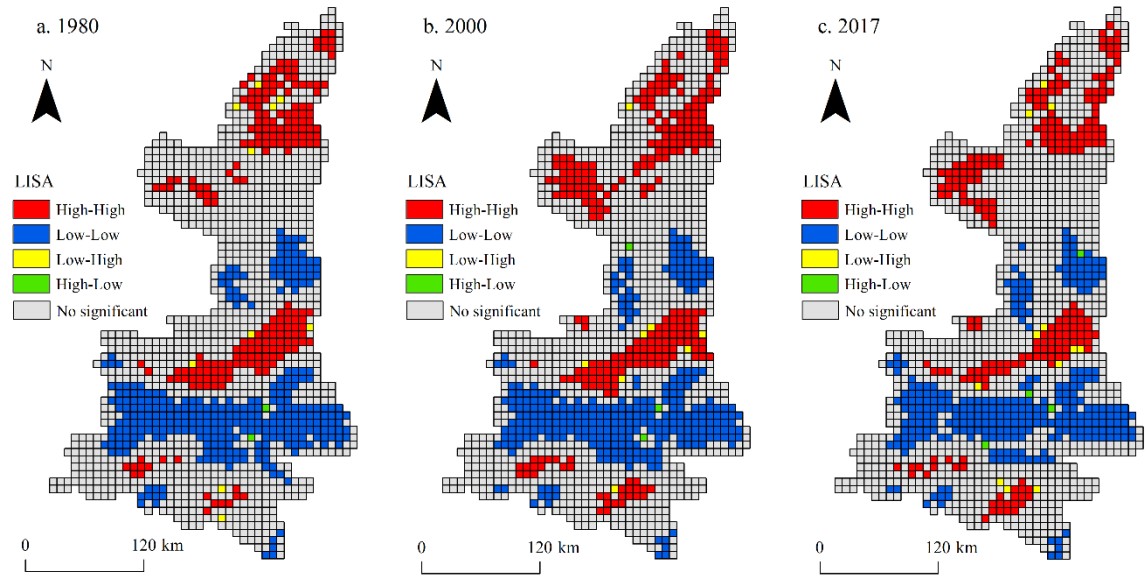

**Figure 3.** Spatial distribution of LISA in Shaanxi Province during 1980–2017.

*4.3. Temporal Variation of Ecological Risk Based on Geomorphologic Regionalization*

The mean value of ecological risk in geomorphologic regionalization was extracted using the Partition Statistical Tool in ArcGIS10.2, and the rate of the temporal variation in ecological risks was calculated (Figure 4). Overall, the changes in ecological risk values were obvious in the different geomorphological regionalizations.

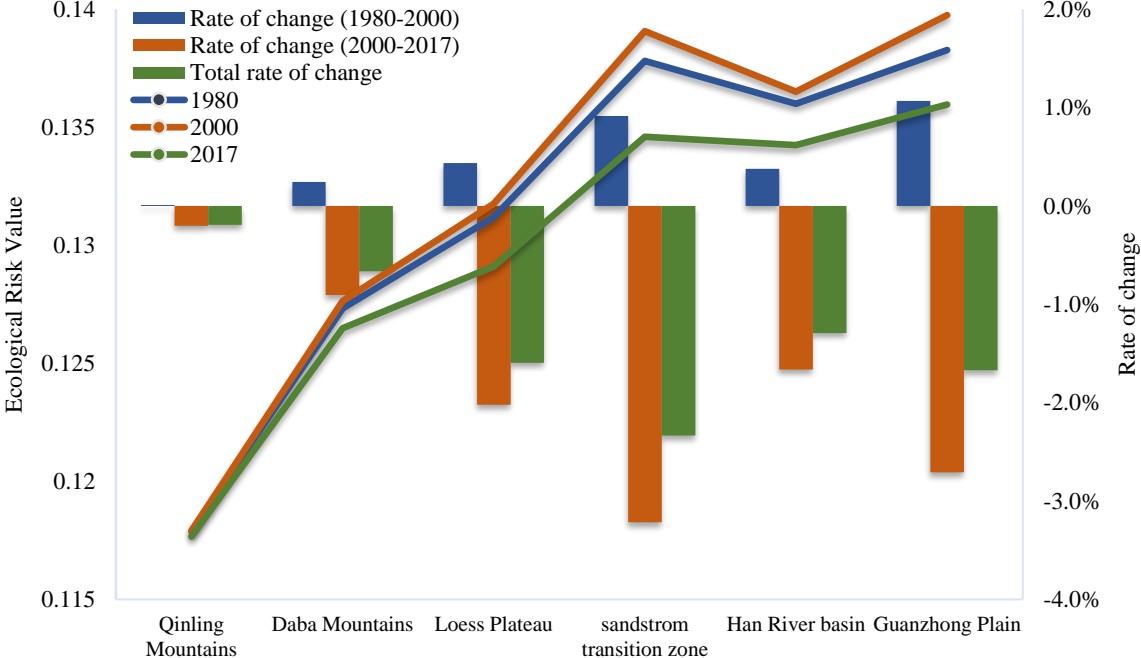

**Figure 4.** Temporal changes in landscape ecological risk based on geomorphological regionalization in Shaanxi Province during 1980–2017.

(1)  The Qinling Mountains (ranked sixth) and the Daba Mountains (ranked fifth) were always low-value areas of ecological risk. The degree of risk evolution of the Daba Mountains was more obvious than that of the Qinling Mountains. Mountainous areas with higher topographical relief

greatly restrict farming behaviors while increasing the expansion cost of urban construction, which results in a low intensity of human disturbance. This scenario led to a high degree of dominance of the regional forest and grassland, forming a relatively closed and safe biological refuge, thus maintaining low ecological risk during the study period.

(2) The internal spatial difference of ecological risk in the Loess Plateau (ranked fourth) was very obvious. The north of the Loess Plateau is the hilly and gully region with higher topographical relief. The disordered landscape structures and broken ecological corridors were formed in this region, resulting in a high level of ecological risk. The forest coverage of the southern part of the Loess Plateau is second only to that of the Qinba Mountains; thus, the ecological risk remained relatively low. Overall, the ecological risk of the Loess Plateau increased from 1980 to 2000, which may reflect the changes in land use types and spatial patterns in the context of agricultural farming and township construction. The rate of decline of ecological risk in the Loess Plateau ranked third in terms of the risk change from 2000 to 2017, which was related to the national policy of returning farmland to forests implemented in 1999. The ecological policy of China increased the area of forests and grassland, reduced the degree of separation of forests and grassland landscapes, greatly reducing the spread trend of ecological risk.

(3) The sandstorm transition zone was located in the northwest of Shaanxi Province (ranked second), where natural vegetation is rare and wind erosion and desertification of soil are very serious. This condition led to a high concentration of unused land, high habitat fragility and high ecological risk. The ecological risk in this region tended to increase from 1980 to 2000. However, a large amount of unused land was converted into forest and grassland after the policy of returning farmland to forest was implemented and this policy enhanced the regional ability to resist sand erosion and effectively reduced the regional habitat sensitivity. The risk reduction rate in this region was the first in terms of the risk change in the corresponding period, with the risk decreasing by 3.22%.

(4) The Han River basin (ranked third) and the Guanzhong Plain (ranked first) were always high-risk areas, but the rates of risk change are fourth and third, respectively. The Guanzhong Plain and Han River basin were the main urban construction areas and important irrigated agricultural areas. Urban construction land and farmland were the most typical man-made landscapes and their land use patterns and boundary fractal characteristics were more marked by human interference, affecting the evolution direction of the landscape pattern. In the early period of disturbance (1980–2000), urban construction land patches were abundant, forming fragmented and disordered landscape areas, which increased the ecological risk. In the later period of interference (2000–2017), the space of construction land expanded rapidly. The construction land with a large area gradually swallowed up the surrounding smaller construction land patches, or many small patches gradually merged to form larger construction land patches, which greatly enhanced the stability of the regional landscape and clearly reduced the ecological risk.

*4.4. Spatial Evolution of Ecological Risk Based on Geomorphologic Regionalization*

The SDE of geomorphological regionalization was generated and the risk gravity was extracted based on the direction distribution tool of ArcGIS10.2. The first-order standard deviation was selected as the model parameter and the ellipse size covered 68% of the risk value (Figure 5). As shown in Figure 5, the shifting direction of ecological risk gravity can be roughly divided into three categories: the southwest, northeast and southeast directions.

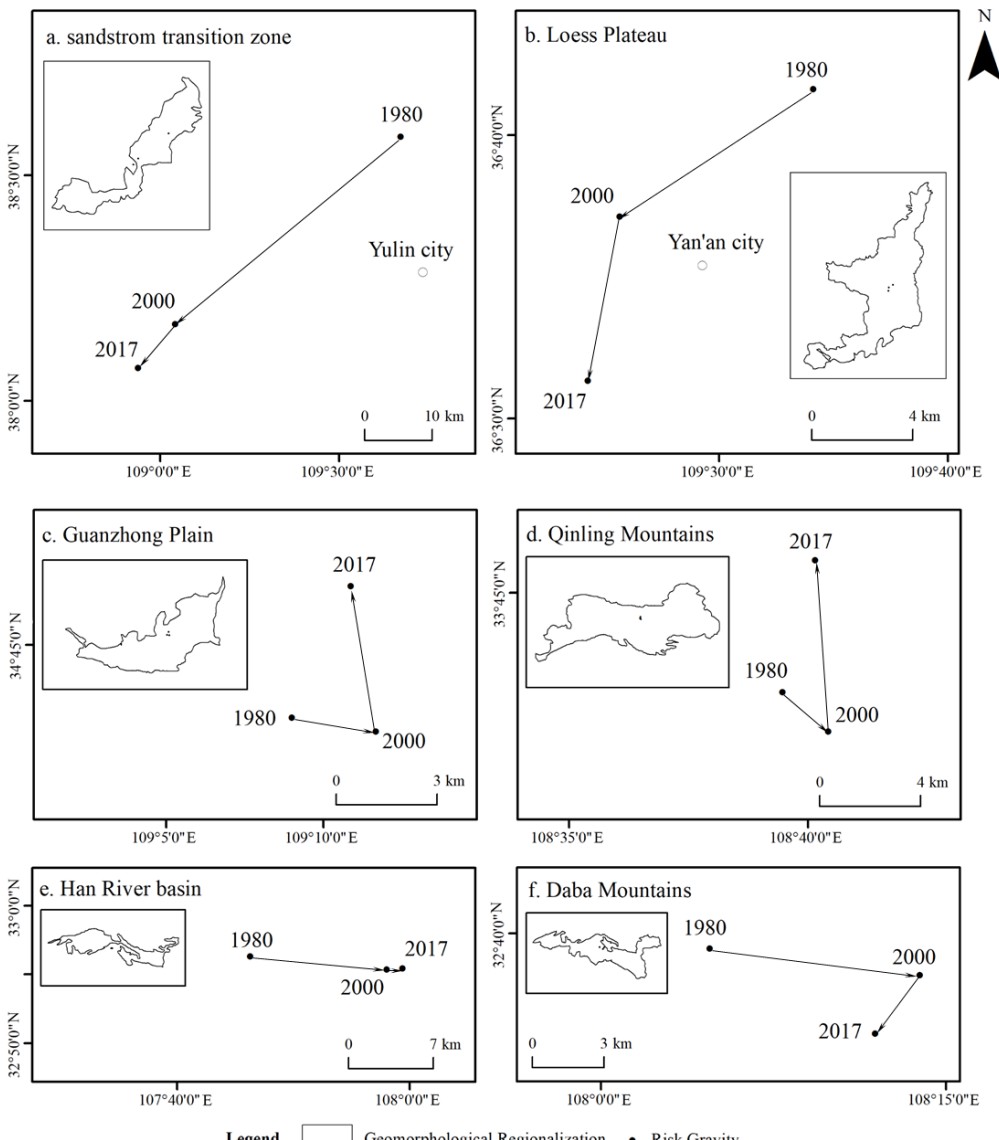

**Figure 5.** Temporal changes in gravity of landscape ecological risk based on geomorphological regionalization in Shaanxi Province during 1980–2017.

(1)  The areas where the risk gravity shifted to the southwest included the sandstorm transition zone and the Loess Plateau. Agricultural production, the policy of returning farmland to forests, and the development of mineral resources should be considered as key factors in the ecological risk changes. This region is important energy accumulation places for coal, oil resources in China. The ecological risk was higher under the influence of a fragile natural basement and excessive energy exploitation. With the transformation of energy structure, the petroleum exploitation in the west of Yulin city strengthened, while the traditional coal cities in the north were at a disadvantage during the industrial transformation, which directly affected the evolution direction of local landscape structure. In addition, as a result of the policy implementation of the three-north shelterbelt construction, a large amount of forest and grassland formed stable habitats in the northern area, which led to the southward shift of risk gravity.

(2)  The areas where the risk gravity shifted to the northeast included the Guanzhong Plain and the Qinling Mountains. Population expansion brought about by urban expansion and agricultural production should be a core factor in risk changes in the Guanzhong Plain. The risk changes

in Qinling Mountains was related to the establishment of protected areas. The Wei River ran roughly from southwest to northeast, which was basically consistent with the direction of the risk gravity shift. The upper reaches of the Wei River have complicated topography, limited farming activities and urban expansion; thus, human activities in the Guanzhong Plain tended to expand to the lower reaches of the Wei River due to improved transportation during the study period. As the national central park, human disturbance acted on the northern and southern slopes of the Qinling Mountains to varying degrees. The northern Qinling Mountains faced a higher degree of human disturbance due to their proximity to the Guanzhong Plain [45], which caused the risk gravity of the Qinling Mountains to move northward.

(3) The areas where the risk gravity shifted to the southeast included the Han River basin and the Daba Mountains. Urban expansion and agricultural production were also considered in risk changes of Han River basin and the risk changes in Daba Mountains were related to the policy of returning farmland to forests. During the research period, the construction land of Hanzhong tended to expand to the downstream regions of the Han River, which destroyed the integrity of the farmland patch. The urban progress of Ankang city was slower than that of Hanzhong city, but the risk increased due to urban expansion. This pattern shifted the risk gravity to the southeast. The increase in the large amounts of forest and grassland areas in the western part of the Daba Mountains further stabilized the ecological process, which significantly reduced the risk in the western part of the Daba Mountains and shifted the risk gravity to the southeast.

## 5. Discussion

### 5.1. Geomorphologic Features, Human Drivers, Pattern Change, Risk Response

The restriction of geomorphologic features on human activities and the natural selectivity of the spatial distribution of land use made human disturbance vary in different geomorphological regionalization scales and the change in related elements, such as landscape structure and habitat fragility, led to the change in ecological risk value in different geomorphological regionalization. In combination with Figure 2a,c, the spatial distribution of human activities under geomorphologic features and the resulting pattern changes and the risk response were discussed by using the south of Shenmu City and the south of Xi'an city, which have obvious risk changes, as an example (Figure 6).

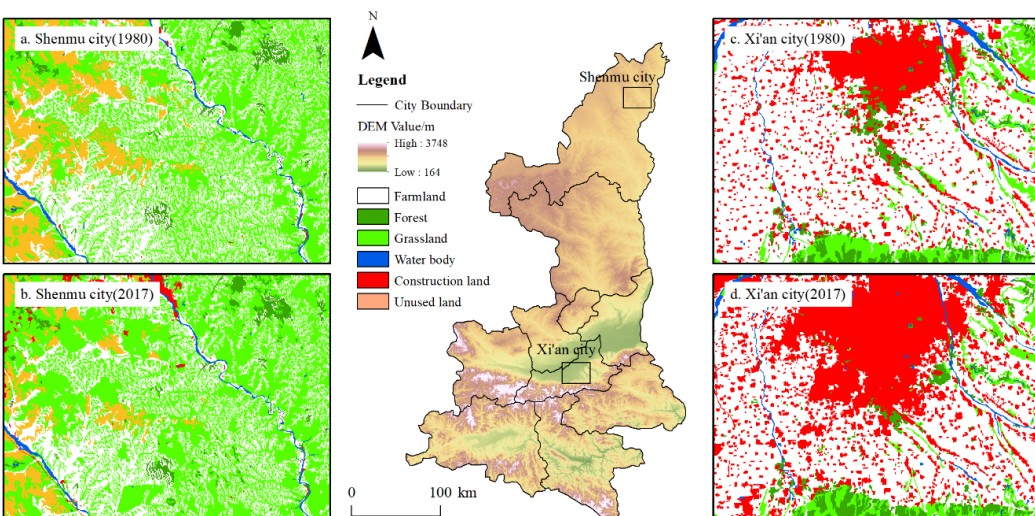

**Figure 6.** The response of landscape ecological risk of typical regions to land use change.

(1) The restricted type: Shenmu City is located in the typical loess hilly gully region with large topographic relief. Human activity was forced to separate because of the mixed terrain, which simultaneously formed a singular and repeated landscape patch. The farmland was located in

terraces or gullies and most of the abandoned farmland was located in the high-slope area. A large amount of farmland and unused land was converted into grassland during the period of farmland regression, which greatly enhanced the dominance of grassland patches and the surrounding stable habitats, benefiting the restoration of regional habitats and effectively reducing ecological risk. In the ecological risk study of the Loess Plateau, Wang et al. believed that the ecological risk of the resettlement area increased first and then decreased from 1995 to 2015 [25], which is similar to the results of this study. However, Ren et al. concluded that the ecological risk on the Loess Plateau has generally increased and declined locally based on comprehensive risk models since 1990 [46]. In short, it is undeniable that the policy of returning farmland to forests has effectively reduced the severity of ecological risks in this region.

(2) The beneficial type: As a typical plain city, the topography of Xi'an was less restrictive to human activities and formed diverse and complex landscape patches. The spatial pattern was dominated by farmland at the beginning of the period and changed to become co-dominated by farmland and construction land at the end of the period. Xi'an city mainly expanded to the southwest during the study period. Chang'an District in the south of Xi'an was dominated by "enclave" patches at the beginning of the period and was gradually "submerged" by spreading expansion. The occupation of farmland by construction land gradually evolved into a more stable large-scale landscape base, thereby increasing the stability of the regional landscape. In the study of the risk of plain cities, Shi et al. believes that the risk of the urban-rural junction is gradually decreasing due to the consolidation of building land patches [47], which is consistent with the characteristics of the ecological risk impact of urban expansion in Xi'an. At the same time, Wang et al. suggested that plain urban agglomerations have always had higher ecological risks due to severe human interference [27]. The risk change of plain cities is closely related to the degree of urban development. Human disturbance can influence the evolution direction of landscape patterns in the plain area, thus leading to an increase or decrease in ecological risk.

In order to better reflect the relationship between pattern changes and ecological risks, Patch Density (PD), Edge Density (ED), Fractal Dimension (FD), CONTAG, DIVISION, and SHDI were used based on the Fragstats4.2 software in typical regions [27,48,49]. The calculation result is shown in Table 3. In Table 3, the ecological risks in typical regions both decreased. At the same time, the PD and DIVISION indexes also decreased, while the CONTAG index increased. This is consistent with the results of related ecological risk studies [49,50]. In contrast, ED, FD, and SHDI have different trends in different regions. These morphological indicators reveal the characteristics of urban morphology [48] and landscape resilience [51], but the accuracy of these indicators for ecological risk characterization needs to be further demonstrated.

**Table 3.** The relationship between morphological indicators and ecological risks in typical regions.

| Typical Regions | Year | Morphological Indicators | | | | | | Ecological Risk |
| --- | --- | --- | --- | --- | --- | --- | --- | --- |
| | | PD | ED | FD | CONTAG | DIVISION | SHDI | |
| Shenmu City | 1980 | 0.6795 | 58.1240 | 1.1270 | 59.0654 | 0.9374 | 1.1314 | 0.1364 |
| | 2017 | 0.6572 | 53.6429 | 1.1257 | 61.0700 | 0.8702 | 1.0774 | 0.1302 |
| Xi'an City | 1980 | 0.7944 | 23.5421 | 1.0529 | 59.3962 | 0.8702 | 1.0646 | 0.1129 |
| | 2017 | 0.7459 | 24.9587 | 1.0593 | 60.3134 | 0.8270 | 1.1187 | 0.1044 |

*5.2. Risk Reduction from the Perspective of Geomorphological Regionalization*

Based on the perspective of geomorphological regionalization, this can provide a basis for regional ecological risk reduction and sustainable development in the context of habitat restoration and pattern optimization.

(1) The Loess Plateau and sandstorm transition zone are ecologically fragile regions in China. In the process of farming and urban construction, the landscape layout and three-dimensional structure

should be reasonably designed according to the original landform, precipitation condition and distribution of forests and rivers [13]. The naturalness of the basement edge of the ecosystem should be maintained while the fragmentation of the landscape should be reduced. The policy of returning farmland to forests should be further implemented, as this policy promotes the restoration of fragile habitats and reduces the sensitivity of regional habitats, thus slowing the spread of ecological risk.

(2) The Guanzhong Plain and Han River basin were the main areas of urban concentration in Shaanxi Province. The terrain struggled to restrict the expansion of Xi'an and other plain cities; thus, the subjective initiative of people can be greatly developed. Efforts should be made to build the ecological space structure of urban agglomerations, and the ecological security pattern was constructed through the optimization and reorganization of the regional ecological matrix, patches and corridors based on the perspective of landscape ecology to effectively reduce ecological risk [52].

(3) The Qinba Mountains are part of a national park and represent an ecological function restriction development zone in China. A new eco-environmental protection regulation on the Qinba Mountains was formulated for vegetation protection, biodiversity maintenance and construction constraints in 2019. Vegetation protection relies on the policy of returning farmland to forests. The expansion of forests and the improvement of connectivity are positive changes in forest landscapes [53], which helps to better exert water conservation and climate regulation. Wildlife habitat reserves have been set up to avoid severe disturbance of biological habitats by human activities. Construction constraints are mainly coping with spatial conflicts with local habitats arising from transportation corridors, rural settlements, and mineral resource development. Landscape management is implemented through corridor design, rural reconstruction, and mineral development governance, thereby reducing ecological risks.

### 5.3. Limitations of Study

The spatiotemporal variation of ecological risk in Shaanxi Province was explored based on the landscape pattern index as a proxy indicator of ecological risk. This method is effective in the absence of long-term regional monitoring data, but it also has some limitations. Using only landscape pattern indicators to characterize ecological risk was incomplete; therefore, indicators of natural disasters and human disturbance risk sources should be given more attention. At the same time, using landscape value [54] and ecosystem services [13] as indicators of the degree of ecological risk loss can more clearly describe the comprehensive loss of landscape and ecosystem under ecological risk stress. The design and implementation of an integrated framework of risk sources and losses is the focus of future ecological risk research. Scale selection is one of the core issues of landscape ecology. Although 12 km was a suitable scale for workload and ecological risk expression, it may not be the optimal scale. Therefore, analysis on the scale effect of landscape ecology on granularity and amplitude will be one of the tasks for the author in the future research. Moran 's *I* and LISA are applied to determine the optimal scale to improve the scientificity of the ecological risk assessment results. The change of ecological risk is affected by many factors and the relationship between these factors and risk should be observed and quantified to explore the main stress factors affecting the change of risk in order to better assist risk management.

### 6. Conclusions

Landscape ERA models based on landscape patterns indexes can effectively reflect the evolution of land use patterns and changes in ecological environments. Depending on the perspective of geomorphological regionalization, the spatial transition of ecological risk can be further clarified. From 1980 to 2017, the landscape ecological risk index of Shaanxi Province was 0.1282, 0.1287 and 0.1273, respectively, with an overall decrease of 0.78%. The Moran's *I* index was 0.7279, 0.7277 and 0.7053, respectively. The ecological risk presented a significant positive correlation, and the degree

of autocorrelation decreased to some extent. The mountains maintained a low ecological risk and a small change rate during the study period. Due to the spatial distribution of building land and arable land, the plain area faced a more serious risk level. Benefiting from the national policy of returning farmland to forests, the ecological risk of the Loess Plateau and sandstorm transition zone greatly reduced and the risk gravity of them gradually retreated southward. The risk gravity of the Guanzhong Plain and Qinling Mountains tended to move northward, while the risk gravity of the Han River basin and the Daba Mountains shifted to the southeast. In the analysis of typical regions, different reasons for the reduction of ecological risks were explored. At the same time, there were different relationships between morphological indicators and risk indexes under different geomorphological features. Therefore, the ability of some morphological indicators to characterize risks should be further verified. Finally, appropriate engineering measures and landscape management for different geomorphological areas were suggested for effective reduction of ecological risks.

**Author Contributions:** D.L., H.C., and H.Z. designed the research and wrote the paper. T.G., Q.S. analyzed the data. All authors have read and agreed to the published version of the manuscript.

**Funding:** This research was funded by National Natural Science Foundation of China (Grant No.41671086 & 41871185) and the APC was funded by Grant No.41671086.

**Acknowledgments:** Supports from the National Natural Science Foundation of China are gratefully acknowledged.

**Conflicts of Interest:** The authors declare no conflict of interest.

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
