# Peer review of "Spatiotemporal Evolution of Landscape Ecological Risk Based on Geomorphological Regionalization during 1980–2017: A Case Study of Shaanxi Province, China"

_sustainability, doi:10.3390/su12030941_

Round 1

Reviewer 1 Report

Dear Authors,

This is a valuable paper concerning a landscape ecological risk however it could need some improvements:

Line 12 Please define land use and land cover and specify to which one you are referring in this paper and why.

Lines 22-32 The conclusions in the abstract should be summed up to the most important information (2-3 sentences maximum). Please do not tell the numbers.

Line 38 Please specify land use or land cover. What do you mean by other land surface elements in this case?

Lines 42-43 Please provide more references if you describe it as popular topic. It is popular indeed – please show it in the sources you quote.

Line 67 Here you call land surface elements a land use or land cover. Please explain.

Line 98 Please change for: in the central part.

Line 167 Please specify if you are referring to landscape types of land use types.

Line 242 “There is…”

Lines 428-433 How is ecological risk managed by planning documents or protection plans (if exist for protection areas)? When the habitat buffer zones were established and how did it influence the ecological risk?

Please discuss the results of other studies in the Discussion section e.g. of the same case study area or similar. The Conclusion section requires a sum up at the end of the paper.

If you find it relevant please refer to the following studies concerning landscape changes, their driving forces and landscape management:

Sylla, M., & Solecka, I. (2019). Highly valued agricultural landscapes and their ecosystem services in the urban-rural fringe–an integrated approach. Journal of Environmental Planning and Management, 1-29.

Solecka, I., Bothmer, D., & Głogowski, A. (2019). Recognizing Landscapes for the Purpose of Sustainable Development—Experiences from Poland. Sustainability, 11(12), 3429.

Solecka, I. (2019). The use of landscape value assessment in spatial planning and sustainable land management—a review. Landscape Research, 44(8), 966-981.

Kind regards,

Reviewer

Reviewer 2 Report

This study presents the spatiotemporal evolution of landscape ecological risk using a landscape ecological risk model. The landscape ecological risk model is developed based on geomorphological regionalization indicators. The spatial and temporal variations of ecological risk can reveal the association between land use changes and the corresponding ecological risk variations.

The study is critical for the morphological studies of land use changes, and the association between human activities and ecological risks. Methods in the study are reasonable to address the problem. Proper data sets are collected and processed. Results and discussion can reflect answers to the primary objectives of the study.

The paper might be improved from following aspects.

(1) Can you add a figure to compare the geomorphological indicators with the areas of land use to reveal the relationship between land use area changes and morphological indicators variations? From my perspective, this figure can be used to directly indicate the major objective of the study. Methods and analysis about investigating area and morphological indicators relationship can refer

https://doi.org/10.1080/13658816.2018.1511793.

(2) In Figure 3, what kind of spatial size or resolution is used for summarizing data to the current grid data? Why this resolution is used? If it is a reasonable resolution? As a recommendation, spatial scale effects on Moran’s I and LISA may be tested to indicate a reasonable scale.

(3) In Figure 5, can you please explain the relationship between risk gravity variations with the potential factors, such as geographical characteristics and socio-economic changes?

(4) Lines 25 – 26: please add years 1980, 2000, and 2017 to the indicators, respectively.

Round 2

Reviewer 2 Report

Thanks for your kind revision. Most of the concerns have been addressed. 

Major comments:

The second comment has not been addressed. Workload is not a reason for selecting a proper size of grids. As a recommendation, can you please compared three optional sizes of grids and compare the size effects?
